# Recent PELE Developments and Applications in Drug Discovery Campaigns

**DOI:** 10.3390/ijms232416090

**Published:** 2022-12-17

**Authors:** Ignasi Puch-Giner, Alexis Molina, Martí Municoy, Carles Pérez, Victor Guallar

**Affiliations:** 1Barcelona Supercomputing Center, Plaça d’Eusebi Güell, 1-3, 08034 Barcelona, Spain; 2Nostrum Biodiscovery S.L., Av. de Josep Tarradellas, 8-10, 3-2, 08029 Barcelona, Spain

**Keywords:** PELE, molecular modelling, virtual screening, machine learning, drug design, CADD

## Abstract

Computer simulation techniques are gaining a central role in molecular pharmacology. Due to several factors, including the significant improvements of traditional molecular modelling, the irruption of machine learning methods, the massive data generation, or the unlimited computational resources through cloud computing, the future of pharmacology seems to go hand in hand with in silico predictions. In this review, we summarize our recent efforts in such a direction, centered on the unconventional Monte Carlo PELE software and on its coupling with machine learning techniques. We also provide new data on combining two recent new techniques, aquaPELE capable of exhaustive water sampling and fragPELE, for fragment growing.

## 1. Introduction

There is little doubt that computational techniques are becoming indispensable when developing molecular therapeutic projects. Whether working on the design of a neoantigen peptide, the engineering of an antibody or an epitope, or when screening a small molecule targeting a protein or RNA/DNA molecule, disrupting a protein–protein interface or enhancing their interaction for degradation, computational techniques are becoming a driving force. Despite recent developments providing models for ADMET and clinical phases [1], most of these modelling efforts focus on the early preclinical phases and, in particular, on the screening and lead optimization stages [2].

Our lab has centered on developing molecular modelling (MM) and bioinformatic techniques for advancing in these preclinical stages; recently, we have also added machine learning (ML) techniques, mostly in combination with MM methods [3]. Our main contribution, the PELE software, could be considered an out-of-the-box Monte Carlo (MC) approach. Instead of performing millions of small fast movements [4], we generate only a few (thousands of) MC steps providing significant conformational rearrangement; in a single step, we might observe a significant ligand translation, coupled with side chain reorganization and backbone displacement [5]. For this, we designed a procedure combining theoretical algorithms with protein structure prediction techniques. As a result, an MC step might take up to a minute to complete, with a full simulation requiring a few hundred steps on several computing cores, each of them performing an individual MC trajectory. These trajectories can be independent or perform a collective effort, using an adaptive scheme [6]. For example, refining a docking pose might require 16–32 computing cores running for 1 to 2 h and provide a fully flexible (all-atom) protein and ligand reorganization [7]. A similar computing effort can also provide growing a fragment into a core, for lead optimization stages, providing an analogous ranking to state-of-the-art FEP + techniques [8].

Results and benchmarks of PELE have been widely introduced in multiple articles [9]. In the CSAR blind international competition, for example, it was recognized as a breakthrough approach, being the main novel technique underlined by the organizers [10]. Such a good performance has driven the transfer of technology to a BSC spin-off company, Nostrum Biodiscovery. Currently, it is being used in prospective projects in multiple countries across Europe, Asia, North America, and Oceania.

In recent years, we have also focused on introducing workflows combining ML models with our MM techniques. This combination aims to bypass applicability domain problems of ML-alone techniques by introducing in silico MM data augmentation [3]. Along this line, novel developments are currently being implemented in screening extra large libraries, such as the REAL one from Enamine, or when combined with generative models.

In this review we will summarize our recent contributions in the drug discovery field, focusing on the potential of our MM methods when used alone or when combined with ML. In the last section, we will also introduce some new developments where we combine two recently developed approaches: aquaPELE [11] and fragPELE [8], for addressing the role of explicit waters when growing ligand fragments.

## 2. Molecular Modelling Advances

Biological systems require molecular models capable not only of describing with high accuracy the structure and energy of molecules but also the dynamics of these systems. The most common strategy to enable movement once molecular energies have been properly described is molecular dynamics (MD), through the integration of Newton’s equations of motion. Many algorithms have been developed using this approach, achieving excellent results and performance, thereby setting it as the standard [12]. However, there are alternative techniques that raise additional lines of action. The most popular alternative is the Monte Carlo method (MC), which relies on a series of artificial perturbation movements randomly applied to the system and, typically, the Metropolis criterion to satisfy the Boltzmann distribution. The advantage of MC over MD is its flexibility when exploring the many degrees of freedom of large systems, resulting in a cheaper method when perturbation algorithms are precisely tailored.

The pioneer of the MC strategy applied to biochemistry is Prof. Williams L. Jorgensen, from Yale University. Along with the popular force field OPLS [13], his group developed BOSS back in the late 1970s, which could be applied to minimize energies of isolated molecules, analyze normal modes, and search conformations [14]. Further development on BOSS led to MCPRO, an MC algorithm specifically designed to study biomolecules in solution. Among its applications, we can highlight the binding mode minimization and refinement [15] and the estimation of binding free energies of protein–ligand complexes upon protein mutations or ligand modifications [4,16,17,18]. Another reference lab is led by Prof. Jonathan Essex at the University of Southampton. Essex explored various applications of MC, of which we highlight the Grand Canonical MC strategy (GCMC), which focused on the sampling of explicit water molecules within biomolecular simulations and the prediction of their thermodynamic properties [19,20], and the Adaptive Sequential MC to enhance the exploration of the conformational space of protein–ligand complexes [21].

As stated in the Introduction section, PELE is our main contribution that complements the aforementioned techniques. Following seminal work from Scheraga combining MC and minimization [22], PELE supports large perturbation algorithms, randomly applied, with tailored relaxation strategies to tolerate big conformational changes while keeping a reasonable Boltzmann acceptance probability. As a result, it is capable of exploring a wide conformational space of large biochemical systems, e.g., to simulate ligand migration pathways within protein cavities, with little computational effort. Examples of these applications include the elucidation of binding mechanisms in nuclear hormone receptors [23], in tyrosinases [24], or in the Ca^2+^ Channel α2δ-1 subunit [7].

The minimization strategy in PELE works along a side chain prediction algorithm that is focused on looking for the best contacts between the protein cavity and the perturbed small molecule, considering its rotatable bonds, as well, thus granting a fair probability of accepting the new state. Consequently, other applications of PELE make use of these exploration capabilities to perform induced-fit docking of small molecules in protein cavities. We can highlight several studies where we map the protein–ligand conformational space with this protocol, such as in the HIV-1 [25] or the soluble epoxide hydrolase [26]. In the HIV-1 case, PELE’s induced fit was challenged against 100 ns molecular dynamics simulations, demonstrating better sampling performance. The hydrolase case strongly challenged the perturbation and side chain prediction algorithms of PELE due to the high flexibility of this protein and the promiscuity of its binding site. Beyond exploration performance, PELE’s algorithms have been positioned as a solution to capture relevant binding modes in quick drug design cycles [27].

Compared to other MC techniques, PELE offers great potential to map medium to large conformational changes. It provides a good compromise between computational resources and exploration of the energy landscape. Global explorations, ligand migration, or local extensive rearrangements are easily handled by routine calculations in PELE [6,28]. On the other hand, PELE cannot perform exhaustive free energy perturbation analysis, such as the ones provided by MPRO or GCMC techniques, due to the lack of resolution when performing a drastic perturbation plus a minimization scheme.

Further development focused on designing external packages to incorporate new capabilities into PELE such as an enhanced sampling strategy, called adaptivePELE [6], and a Markov State Model to estimate absolute binding free energies, named MSM-PELE [29]. The first solution enabled faster exploration protocols introducing biased sampling that could boost exploration by up to 10 times the standard code. This strategy has been helpful to determine ligand binding pathways, for example, in the nuclear hormone receptor [30] and in the vanillyl, alcohol oxidase [31]. Additionally, PELE’s sampling could handle global explorations to identify binding sites around the protein surface as in this study where ligand binding interfaces for protein–protein inhibition in the flu virus hemagglutinin were identified [32]. The second package, despite the challenge it represents, offers a promising approach to the determination of absolute binding free energies, especially in systems with non-occluded binding sites such as urokinase or plasmin receptors where binding events can easily happen.

Latest developments focused not only on enhancing applications on global explorations but also on improving predictions in local protein pockets. To better determine affinity changes upon ligand changes, assisting hit-to-lead campaigns, we developed fragPELE [8]. It is a fragment-based alchemical growth procedure that relies on a series of consecutive PELE simulations capable of incorporating additional molecular fragments onto a structural scaffold (referred to as the core, already bound into the protein). Thus, the effects of the new fragment are gradually introduced into the model through the modification of the forcefield parameters. As a result, fragPELE executions allow a quick structural and energetic comparison of the ligand upon fragment addition, including protein flexibility such as side chain reallocation.

Complementing the fragPELE approach, we recently developed a hybrid method to treat solvent effects in PELE to further improve accuracy. Standard PELE uses implicit solvent models to account for the effects of water. To benefit from the speed of implicit models and, at the same time, explicitly introduce and sample water contributions ignored by implicit approaches, we developed aquaPELE [11]. It introduces an extra perturbation algorithm specifically designed to sample explicit water molecules within a pocket. Thus, it can handle explicit water molecules in combination with the surface generalized Born solvent model (with a variable dielectric implementation and a nonpolar term), which is used to consider solvent effects in the rest of the system. This extra perturbation step allows several water molecules to adapt to the ligand, enabling the detection of hydrogen bonds or water displacement upon ligand entrance. As a result, aquaPELE offers a good solution to deal with systems where water plays important roles in ligand recognition. It is also an interesting plugging for fragPELE as it can be used to detect which fragment candidates are more suitable for water displacement. The combination of both techniques is discussed below in more detail.

## 3. Combining ML and MM

Applicability domain, quality, and optimal performance of stand-alone machine learning techniques are highly dependent on available data. When following the ML panels in recent conferences, such as the recent Discovery on Target (Boston, October 2022), for example, we find growing doubts about the (possibly oversold) potential of such techniques in today’s real prospective implementations. As an alternative, researchers might focus on using MM as a data augmentation tool and combine its outcomes with data-intensive algorithms and frameworks, such as generative models or big data management systems. Docking techniques, MC simulations, or MD are powerful tools to generate affinity predictions and quickly create large datasets of information on protein–ligand complexes, thus, the synergy between MM methodologies and paradigms that benefit from large amounts of data is obvious, which led the field to provide many examples in the literature of the application of MM combined with ML to perform specific tasks. In 2017, Ash and Fourches applied descriptors from MD simulations to find binders on ERK2 [33]. In the same year, Ding et al. employed molecular dynamics fingerprints (MDFP) [34] to predict toxicity for the hERG channel [35]. Jumping to docking, some groups trained ML scoring functions on docked complexes to improve the accuracy of their predictions [36,37], and, following a similar trend, Sanner et al. applied ML to classify peptide affinities by using docking complexes [38].

When using such an approach, one should consider the computational cost associated. MD is a powerful tool to augment data, but its high computational cost would restrict its use to a limited amount of compounds, which is not ideal for complex analysis or when aiming for an iterative self-learning implementation. In the latter scenario, cheaper methods such as docking or MC simulations are more adequate options. The following subsections describe three different approaches where we use such cheaper methods involving, in addition, different ML categories.

### 3.1. MM Data Augmentation Enhances ML Downstream Tasks

We have implemented new approaches to mix MM with ML methods to optimize the top rankings in a hierarchical virtual screening campaign. In a recent study, we ran short induced-fit PELE simulations on top of docking results to amplify the data, analyze, and then collect relevant (heterogeneous) binding descriptors to assess the ligand’s potency. This information was mixed with pure ligand-based properties in a multiple-source dataset (Figure 1) to train an ML classifier capable of differentiating high- from low-active hits. The validation results have shown that the combination of all features in an ML model describes the compound’s activity better than using individual descriptors (or those coming solely from one simulation technique). The resulting technique was used in a real-world screening collaboration with the pharmaceutical company Almirall, where we refined 785 compounds from a primary virtual screening in 6 days (using 512 computing threads in parallel), selecting 23 for experimental testing. In the end, two hits were unearthed, one in the nanomolar range activity [3].

### 3.2. Directed Generation of New Chemical Entities

Turning to more recent ongoing work, one of the most promising applications that arises from the combination of MM and ML is the generation of novel compounds starting from an initial set of molecules of known relevance or special interest. Generative artificial intelligence has experienced an upsurge of different techniques in the last few years, providing several other fields with a solid ground of frameworks and enabling its use in combination with well-established methodologies. A great example of this synergy is the use of generative models along with MM methods to generate relevant molecules compliant with a specific set of properties, such as drug-likeness, synthetic accessibility, molecular weight, or solubility. This process alone does provide value to the problem of exploring unknown instances of the chemical space but falls short when aiming for up-to-par compounds in terms of affinity for a target of interest.

Our approach (Figure 2) consists of combining several rounds of generation within an active learning paradigm, which ensures that the model learns not only from the initial known set of molecules but also from the best ones ranked under the scope of the chosen metrics. After several rounds of generation, once enough chemical diversity has been built, generated compounds undergo docking or PELE short simulations with a chosen protocol previously established (such as the ones defined in Borrelli et al. [28]). This last step gives a meaningful approximation of the affinity, a parameter utilized as a threshold for enriching again the initial set of compounds, which serves again as an input set for the model as a new initial set.

Iteratively, QED, SA, and affinity distributions shift towards better scores or, in a worst-case scenario, remain stagnant at initial distribution values, but the process enhances chemical diversity.

### 3.3. Screening of Ultra-Large Databases

An analogous approach, iterating between ML and MM methodologies, is also being used to screen ultra-large databases. The ENAMINE Real database is one of the largest collections of commercially available compounds and also one of the most used for screening campaigns. With 4.5B compounds, its use changes the paradigm under which common search and retrieval algorithms work efficiently and under reasonable time and computational resources.

We enable fast searches of molecules in ultra-large libraries of compounds by computing MinHash fingerprints [39] for the whole set in an HPC regime along with due compressed storage of such data. To target those regions of the space populated with potential inhibitors, an evenly distributed sampling of the database is performed upon a clusterized version of the collections. Although pre-computation of all the necessary descriptions and clusterization of the totality of the set is computationally consuming and technically challenging, it is necessary to provide a faithful representation of the chemical matter.

With clustering as a starting point, a certain number of molecules on the verge of a few million are selected as the initial batch. Those undergo HTVS followed by more refined protocols such as Glide docking [40,41,42] and PELE. The scoring power of molecular mechanics algorithms sets the ground for focusing the search on relevant parts of the space by using the best-ranked compounds and searching back to the library converting retrieved interesting molecules into subjects of the queries. Iteratively executing this process, the queries retrieve compounds more potent as more relevant regions of the chemical extent are explored exhaustively.

## 4. Combining fragPELE and aquaPELE

The fragPELE method, as well as docking techniques in general, is highly dependent on the hydration conditions of the protein’s binding site [8]; simulations in hydrated binding sites have poorer structural and energetic predictions than in hydrophobic ones. We also presented the aquaPELE method [11], aimed at addressing the role of explicit waters. These two developments made us hypothesize that the combination of fragPELE and aquaPELE could significantly improve the structural and energetic predictions when growing ligands in hydrophilic active sites. In this section, we present the results of such a combination. We organized our study in (1) structural analysis, where the aim was to predict the location of crystal water molecules both before and after the fragment growing; (2) energetic analysis, where we focused on the prediction of the free binding energy changes of a series of congeneric ligands.

In the structural validation, we assembled a benchmark of eight ligand scaffolds [43,44,45,46,47] where we grew fragments to obtain ten final fully grown ligands (Appendix A). For each of them, we performed two simulations: (1) a simulation of the ligand scaffold alone with a few explicit and fixed water molecules while the rest were perturbable (Appendix A), and (2) a second simulation where we made the fragment growing with the same configuration of waters. The system preparation and the protocol used for all the simulations can be found in Appendix A. The objective of this study was to see whether the hydration sites of the crystal were predicted before and after the fragment growth. To assess it, we applied a clustering strategy to the water positions reached throughout the simulation (Appendix A). We tracked the perturbable water’s oxygen distance to the crystallographic position and evaluated the cluster’s density. These results are summarized in Table 1. In general, the water molecules were properly located, both in the scaffold simulation with aquaPELE and in the growing one using the aqua + fragPELE combination. In aquaPELE simulations with the scaffold, the distances from the predicted hydration sites to the crystal remained below 1 Å in ten out of eighteen and below 2.5 Å in all of them. The growth of the fragment caused the water migration only in the expected cases. For all the systems where the fragment displaced a water molecule, our simulations predicted the decrease in density (or displacement) for that water cluster. This indicates that the water reduces the frequency of the location due to the effect of the corresponding fragment. A clear example is the results obtained in BRD4 (PDB file 5I80), where both waters completely disappear after adding the fragment (Figure 3). Finally, the control cases where the fragment growing did not displace the water also met our expectations, having either nearly zero or positive values in the density change.

For the energetic benchmark, we used three series of congeneric ligands [43,44] (Appendix A). In each series, we had an initial scaffold from which we grew some fragments to obtain a series of fully grown ligands with their respective associated binding free energy (Appendix A). The objective was to see how the presence of explicit water molecules affected the energetic result. That is why we came up with three different conditions: (1) the water condition, with perturbable and fixed explicit water molecules (abbreviation: W); (2) the fixed waters condition, only with fixed water molecules (abbreviation: FW); and (3) the no-waters condition, without explicit waters and only featured the implicit solvent (abbreviation: NW).

We computed six different scoring criteria based on PELE’s interaction energies (Appendix A) and applied all of them to all the complexes. In addition, for each congeneric series, the correlation analysis between the predicted and the experimentally determined affinities were obtained for each of the three different water protocols previously noted (NW, F, and W). Thus, for the three series and three water conditions, we calculated the average and the standard deviation for each of the six scoring functions. We then did the average correlations between the three congeneric series to generate Figure 4. As expected, we observed that the best correlations involve the W protocol followed by FW and NW (Appendix A). The protocol with fixed and perturbable water molecules had the best results across all scoring methods. It is also the more stable protocol, as the average standard deviation of the averaged points was $\sigma_{W}$ = 0.03 ($\sigma_{FW}$ = 0.06 > $\sigma_{NW}$ = 0.04). As for the different scoring methods, we do not see any clear trend when comparing the three different water models. In any case, for the W protocol, all scoring methods give correlations in the 0.72–0.76 range, indicating its high predictive power.

## 5. Conclusions

We have summarized in this work our recent contributions on developing and applying computational techniques in the early screening and optimization phases of drug design. From an early emphasis on molecular modelling, we are quickly moving to prioritize the combination of ML and MM techniques. Such a combination might be the best approach for the coming years until data and technology allow ML-alone methods to become predominant. Still, the complexity of a robust modelling prediction, such as shown here when comparing implicit and explicit solvent predictions, might delay such an overthrown point.

## Figures and Tables

**Figure 1 ijms-23-16090-f001:**
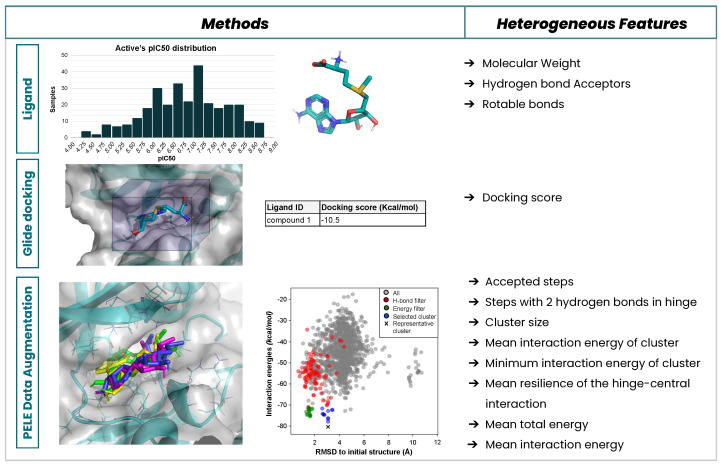
Heterogeneous data extraction and feature build-up in our recent prospective study for the Almirall company. Different levels of model description and molecular modelling provided several features for training a model with a set of experimentally validated compounds.

**Figure 2 ijms-23-16090-f002:**
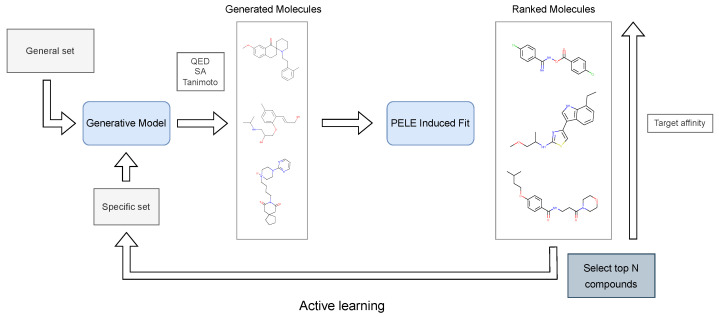
Active learning schema depicting the process used for iterative generation and enhancement of potential inhibitors.

**Figure 3 ijms-23-16090-f003:**
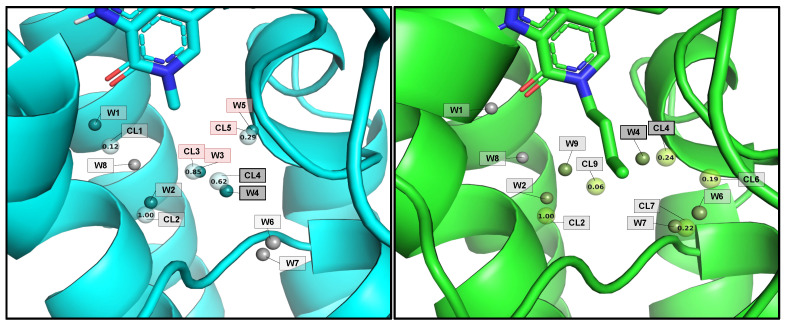
Demonstration of the fragment growing effect in BRD4. In both images, the solid and the transparent spheres are crystallographic and water clusters, respectively. On the left-hand side, we have the results of the aquaPELE simulation. Highlighted we can see W3, CL3 and W5, CL5 almost overlapping. These waters correspond to A319 and A336, respectively, which according to Appendix A are the waters to be displaced upon fragment growth. On the right-hand side, we can see that the water clusters from the frag + aquaPELE simulation disappear, as we show in Table 1.

**Figure 4 ijms-23-16090-f004:**
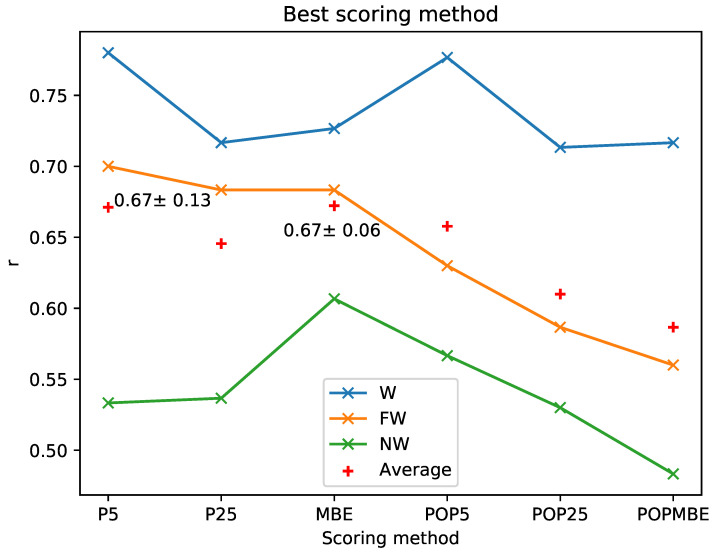
Average r of the three values obtained for each series of ligands for every scoring method. In the case of the fifth percentile and the mean binding energy (P5, MBE), the average with its standard deviation is indicated. No error bars are shown to simplify the visualization of the diagram.

**Table 1 ijms-23-16090-t001:** Structural information of the results obtained for all the important water clusters in the simulated systems. The checkmark indicates that the distance between the detected water cluster’s oxygen and the crystallographic position of the water’s oxygen ($r_{cc}$) is less than 1 $\overset{˚}{A}$. When one density extends over two different water IDs (i.e., HSP90(1), A249 and A286: ρ=1.0) means that only one cluster was detected between the crystallographic positions of the original waters.

Systems	Scaffold					Growing	
	PDB Scaffold		Water ID	ρ	$\Delta r_{\mathrm{cc}}<1\overset{˚}{A}$	PDB Grown	Δρ
HSP90 (1)	3RLQ		A249	1.0	1.84	3RLR	−0.45
			A286		1.25		-
HSP90 (2)	2XAB		A2246	1.0	✓	2XJG	−1.0
			A2115	0.60	✓		+0.05
BRD4	5I80		A319	0.85	✓	5I88	−0.85
			A336	0.29	✓		−0.29
TAF1	5I29		A1891	0.16	✓	5I1Q	A1891: −0.13
							A1860: −0.79
			A1860	0.84	✓	6BQD	A1891: −0.16
							A1860: −0.71
SiaP WT	2V4C		A2346	0.07	✓	3B50	−0.07
CHK1	2C3L		A2056	0.02	✓	2C3K	−0.02
			A2127	1.0	1.44		−0.91
			A2052	0.07	1.52		−0.07
			A2043	0.02	1.98		−0.02
Control	HSP90 (1)	3RLQ	A249	1.0	1.84	-	+0.03
			A286		1.25		0.0
	HSP90 (2)	-	A1	1.0	✓	3RLP	−0.03
			A3	0.8	✓		+0.20

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
