# Peer review of "Recent PELE Developments and Applications in Drug Discovery Campaigns"

_ijms, 2022, doi:10.3390/ijms232416090_

Round 1

Reviewer 1 Report

In this manuscript, the authors present a brief review and continued development of the protein structure modeling software PELE developed in the lab of the corresponding author. The presentation of concepts in the manuscript provides a compact summary of PELE’s contributions to first principles based molecular modeling (MM) and small molecule docking in the context of drug discovery efforts. However the presentation lacks clarity when talking about the combination of MM with machine learning (ML), especially in Section 4, where the mentioned applications appear somewhat disconnected. The first application, namely MM guided data augmentation over MD simulations for downstream ML tasks, is basically an example of computational pipeline development. The second application involving active learning for learning chemical information space in the context of small molecule docking is an example of ML assisted method development. The third application i.e. screening large compound databases is an example of big data management. Modern ML is often (ab-)used as a broad umbrella term which includes not only statistical algorithm development but also exploratory data science and big data software management. I suggest that the authors not push themselves into this trap of umbrella usage of terminology and instead make Section 4 sharper and more precise by categorizing the different examples according to the nature of the corresponding application. 

I also find the title too broad. Why name a paper differently than the intended story? If this is a review and update on the latest features of the PELE framework, why not name it just that? For example, one can argue that drug discovery in the most general sense includes biologics (peptides, antibodies, etc) based drugs as well and yet the major applications of PELE cited and illustrated in this work all focus on small molecules. Please make sure that the title reflects the contents of the work. 

Line 27-28: If the authors simply say that each MC step can take a long time to compute (~ 1 min), then it naturally begs the question of how efficient the entire algorithm is even after taking a smaller number of overall steps compared to usual translation / rotation MC moves. Please present / cite some measure of relative efficiency over more traditional MC algorithms for molecular simulations. 

Line: 79: Spelling mistake: Scheraga 

Author Response

Dear reviewers and editors,

Many thanks for your positive review, which has further enhanced the manuscript.  Please find a detailed answer to all reviewers below. Also a version with all changes highlighted in blue has been compiled.

Sincerely

Victor Guallar

ICREA Professor

Barcelona Supercomputing Center.

Reviewer 1:

However the presentation lacks clarity when talking about the combination of MM with machine learning (ML), especially in Section 4, where the mentioned applications appear somewhat disconnected. The first application, namely MM guided data augmentation over MD simulations for downstream ML tasks, is basically an example of computational pipeline development. The second application involving active learning for learning chemical information space in the context of small molecule docking is an example of ML assisted method development. The third application i.e. screening large compound databases is an example of big data management. Modern ML is often (ab-)used as a broad umbrella term which includes not only statistical algorithm development but also exploratory data science and big data software management. I suggest that the authors not push themselves into this trap of umbrella usage of terminology and instead make Section 4 sharper and more precise by categorizing the different examples according to the nature of the corresponding application. 

Reply: This is a good point by the reviewer. Being somehow new to the ML field we take note of it! Accordingly we have categorized the section. Thank you. 

I also find the title too broad. Why name a paper differently than the intended story? If this is a review and update on the latest features of the PELE framework, why not name it just that? For example, one can argue that drug discovery in the most general sense includes biologics (peptides, antibodies, etc) based drugs as well and yet the major applications of PELE cited and illustrated in this work all focus on small molecules. Please make sure that the title reflects the contents of the work.

Reply : Title changed to: Recent PELE Developments and Applications in Drug Discovery Campaigns”

Line 27-28: If the authors simply say that each MC step can take a long time to compute (~ 1 min), then it naturally begs the question of how efficient the entire algorithm is even after taking a smaller number of overall steps compared to usual translation / rotation MC moves. Please present / cite some measure of relative efficiency over more traditional MC algorithms for molecular simulations. 

Reply: A single PELE MC step can involve quite a large conformational change, including ligand translation by few angstroms, side chain rotamer change and slight backbone displacement (~1 angstrom). This is dictated by different parameters, such as the ligand translation one. In a few hundreds unbiased PELE steps, for example, we can observe a full migration of a fatty acid escaping a fatty acid binding protein [doi: 10.1021/ct0501811], or how tiotropium binds into the muscarinic receptor from outside the solvent [doi: 10.1038/s41598-017-08445-5]. It is difficult to compare with other MC methods since I am afraid no other MC technique can produce that kind of sampling without strong biased potentials. On the other site, traditional MC methods might be more precise to obtain free energy changes in relatively small perturbations (where PELE’s resolution might not be sufficient) or in absolute binding affinities; I still have to see a convincing MC technique, however, producing such a deltaG analysis in a broad set and with meaningful changes. We have added some more details in the manuscript addressing these differences.

“Compared to other MC techniques, PELE offers great potential to map medium to large conformational changes. It provides  a good compromise between computational resources and exploration of the energy landscape. Global explorations, ligand migration or local extensive rearrangements are easily handled by routine calculations in PELE [6,28]. On the other hand, PELE cannot perform exhaustive free energy perturbation analysis, such as the ones provided by MPRO or GCMC techniques, due to the lack of resolution when performing a drastic perturbation plus a minimization scheme.“

Line: 79: Spelling mistake: Scheraga 

Reply: Changed

Reviewer 2:

Comments to the author(s):

1. Figure 1 and Figure S1 are not clear, text font size is small. I would suggest the authors should use high-resolution images.

Reply: Figures have been improved in text size, etc. Thank you

2. The authors mention that generated compounds undergo docking or PELE short simulations with a chosen protocol previously established. The authors should cite appropriate references for the same. 

Reply: Yes,  thank you. A citation has been added in this context

3. The authors should consider including one more case study to evaluate the prediction efficiency provided by PELE. The authors should mention how their techniques are efficient/different than the other well-known similar drug design tools.

Reply: In response to this question and the last one of reviewer 1, we have added an additional paragraph. Also a better description of our HIV paper, where results were compared with exhaustive MD.

We are afraid that we cannot give more details on prospective studies (success rate…), due to confidentiality. The one for Almirall is one of the few ones financed by public funds where we can disclose some results. In addition, due to the industrial licenses being issued at Nostrum Biodiscovery,  we are banned from publishing any comparison with our vendors, such as Schrodinger (Glide, Desmond, Macromodel, etc.). Still, in the added paragraphs we included a couple of previous references where we performed general comparison (before being banned).

Chapters added and modified introducing comparison:

“...In the HIV-1 case, PELE's induced fit was challenged against 100 ns molecular dynamics simulations, demonstrating better sampling performance. The hydrolase case strongly challenged the perturbation and side chain prediction algorithms of PELE due to the high flexibility of this protein and the promiscuity of its binding site. Beyond exploration performance, PELE's algorithms have been positioned as a solution to capture relevant binding modes in quick drug design cycles [27].

…Compared to other MC techniques, PELE offers great potential to map medium to large conformational changes. It provides  a good compromise between computational resources and exploration of the energy landscape. Global explorations, ligand migration or local extensive rearrangements are easily handled by routine calculations in PELE [6,28]. On the other hand, PELE cannot perform exhaustive free energy perturbation analysis, such as the ones provided by MPRO or GCMC techniques, due to the lack of resolution when performing a drastic perturbation plus a minimization scheme.”

Reviewer 2 Report

Comments to the author(s):

 This manuscript reports the recent contributions of Monte Carlo PELE software and its coupling with machine learning techniques in the early screening and optimization phases of drug design. The organization and writing for this manuscript is clear, but my comments/suggestions should be addressed.

  1. Figure 1 and Figure S1 are not clear, text font size is small. I would suggest the authors should use high-resolution images.
  2. The authors mention that generated compounds undergo docking or PELE short simulations with a chosen protocol previously established. The authors should cite appropriate references for the same. 
  3. The authors should consider including one more case study to evaluate the prediction efficiency provided by PELE. The authors should mention how their techniques are efficient/different than the other well-known similar drug design tools.

Author Response

(The authors gave the same response as above.)
